# Non-Specific Effects of Prepartum Vaccination on Uterine Health and Fertility: A Retrospective Study on Periparturient Dairy Cows

**DOI:** 10.3390/ani15172589

**Published:** 2025-09-03

**Authors:** Caroline Kuhn, Holm Zerbe, Hans-Joachim Schuberth, Anke Römer, Debby Kraatz-van Egmond, Claudia Wesenauer, Martina Resch, Alexander Stoll, Yury Zablotski

**Affiliations:** 1Clinic for Ruminants with Ambulatory and Herd Health Services, Centre for Clinical Veterinary Medicine, Ludwig-Maximilians-University, 80539 Munich, Germany; h.zerbe@lmu.de (H.Z.);; 2Institute for Immunology, University of Veterinary Medicine, 30559 Hannover, Germany; hans-joachim.schuberth@tiho-hannover.de; 3Mecklenburg-Vorpommern Research Centre for Agriculture and Fisheries, Institute of Livestock Farming, 18196 Dummerstorf, Germany; a.roemer@lfa.mvnet.de; 4RinderAllianz GmbH, 17348 Woldegk, Germany; 5Intervet Deutschland GmbH, 85716 Unterschleissheim, Germany

**Keywords:** transition cow, vaccination, trained immunity, cattle, bovine metritis, epidemiology

## Abstract

Prepartum vaccination, given to cows during late pregnancy protects calves during their first weeks of life. While the benefits for calves are well established, less is known about potential effects on the cows themselves. Recent research suggests that vaccines may influence not only protection against specific diseases but also overall health and productivity, through so-called non-specific effects (NSEs). This study analyzed data from 20 German dairy farms to examine the association between prepartum vaccination and uterine health and reproductive performance. The results showed that herd management and milk yields had the strongest overall impact. However, cows receiving non-live (inactivated) vaccines were more likely to develop uterine diseases and showed reduced fertility compared to unvaccinated cows. In contrast, certain vaccination timings and the use of live vaccine components were associated with improved fertility. These findings point to possible NSEs of prepartum vaccination on reproductive health. Further research is needed to clarify the underlying mechanisms and to optimize vaccine timing and composition.

## 1. Introduction

Non-specific effects (NSEs) of vaccination beyond an antigen-specific induction of antibodies and effector T cells can be related to both adaptive and innate immune mechanisms, such as heterologous T cell immunity and trained immunity [1]. The extent and nature of NSEs are intensely debated, and both beneficial and adverse effects are reported [2]. Epidemiological studies of human vaccines have been well elaborated [1], and evidence of NSE has been shown, especially for live-attenuated vaccines, such as Bacille Calmette–Guerin (BCG), measles, oral polio, and vaccinia [1]. In veterinary medicine, research is still lacking, especially for non-live vaccines [2]. Recently, the hypothesis that live vaccines have beneficial and non-live vaccines have detrimental NSEs [3] has been confirmed in a systematic review [4]. In cattle, NSEs have been demonstrated in calves, where circulating myeloid cells displayed a trained functional phenotype after experimental BCG vaccination [5]. For non-live vaccines, an altered gene expression of circulating immune cells has been demonstrated in cows [6,7]. In a recent study, prepartum vaccination against neonatal calf diarrhea (NCD) had no significant effects on mammary health and milk yield. However, the study did not address the influence of live and non-live components [8].

Prepartum vaccination of cows are used to protect calves from infectious diseases during the first weeks of life. It is considered safe for the dam [9,10]. Although vaccination of pregnant women during the COVID-19 pandemic demonstrated no overall adverse effects, the most opportune timing of vaccination to elicit an optimal immune response in the mother to benefit the neonate remains unclear [11]. The effect of timing of vaccination during pregnancy remains largely unexplored. However, findings from epidemiological studies involving children who have been vaccinated against measles and with a BCG-vaccine indicate that the timing of vaccinations can have a substantial impact on NSEs [12,13]. In dairy cows, the timing of prepartum vaccination with two non-live vaccines against NCD and mastitis and its relation to pen changes and the acidogenic diet has been shown to affect lying time, metabolic profile, and immunoglobulins [14,15]. Prepartum vaccination of cows takes place before or directly in the transition period, defined as three weeks before parturition to three weeks after parturition [16]. Thus, the timing of prepartum vaccination may be an issue, since the transition from a lactating cow to a dry cow is associated with altered immune cell functions [17,18].

Depending on the time when pregnant cow immune cells change their functionality before parturition, initial innate immune responses to vaccinations may differ, and hence the induced NSE after vaccination. Fine-tuned innate immune mechanisms are already important in the prepartum period [19], and dysregulated responses extend into the postpartum period, co-determining whether uterine contamination proceeds to recovery or to an infectious disease. The altered immune system during the transition period is considered a key factor in the development of postpartum uterine infections and subsequent fertility problems [20,21,22,23]. A retained placenta is defined as the failure to pass the fetal membranes within 24 h. The remaining membranes are usually noticed within the first days postpartum [19]. While most metritis cases occur within the first ten days postpartum, a cut-off date between metritis and endometritis can be made on day 21 postpartum [24]. Later in the cows’ production cycles, the immune system and the presence of previous uterine disease co-determine the cows’ fertility [25].

We hypothesize that prepartum vaccination modulates immune mechanisms in a way that affects uterine disease susceptibility. The objective of this retrospective study was to analyze the associations between the prepartum vaccination against NCD, vaccination timing and uterine health and fertility in the periparturient cow.

## 2. Materials and Methods

### 2.1. Data Collection

A total of 20 dairy farms enrolled in the RinderAllianz test-herd program were selected for this study, including ten farms that carried out prepartum vaccination against NCD in June 2021, and ten farms that did not perform prepartum vaccination. Milk yield was balanced in order to attain five high yielding and five low yielding farms in each vaccination group with a cut-off value of 11,000 kg mean energy-corrected milk yield over 305 days of lactation (ECM 305).

Data was extracted from the herd management programs, comprising reproduction data of 148,268 lactations, milk data from 1,561,273 recordings, health documentation comprising 1,298,703 diagnoses and holding registers for 73,378 dairy cows from 22 herds across the 20 farms between January 2007 and September 2020. The large size of this dataset revealed sufficient statistical power, making a priori power analysis unnecessary. Additionally, an on-farm survey was conducted on all farms between October 2021 and August 2022 to record the local conditions and management practices. To this end, a questionnaire on vaccination history, dry-off management, housing system, health management, monitoring during birth, milking and colostrum management, hygiene and feeding management was designed. An English translation is provided in Appendix A. To ensure consistency in data collection, the same surveyor administered all questionnaires. One farm was surveyed remotely due to pandemic-related precautionary measures; all remaining farms were visited on-site, enabling a more comprehensive understanding of the farm environment. Farmers provided surveyors with full access to all pertinent areas of the facility, including animal housing at all production stages, as well as milking systems and calving areas, which were subject to particularly close inspection. In addition to data collection, these visits allowed surveyors to verify the absence of any overt deficiencies in management or hygiene. Furthermore, the attending veterinarians of the farms were consulted to enhance the evaluation of herd management. When scheduling allowed, these veterinarians participated directly in the on-site visits; in other instances, they were consulted separately.

### 2.2. Data Pre-Processing and Variable Definition

On-site survey data were manually transferred from paper forms to ExcelTM-spreadsheets. R version 4.3.1 software was used to match the questionnaire with data from the herd management programs. Information from the questionnaires revealed that five herds were consistently vaccinated, eight herds remained unvaccinated throughout the study period, and eight herds were intermittently vaccinated with the prepartum vaccine against NCD during the entire study period between 2007 and 2020. The decision to vaccinate was made at the herd level, with timing ranging from eight to 2.5 weeks before calving. Within each herd, the vaccination protocol was consistent for all cows, although in some cases heifers were vaccinated at a later time. While vaccination protocols differed between herds, all farms followed the manufacturer’s guidelines. Due to the timely fragmentation of the vaccination protocols, it was decided to perform the statistical analysis at the cow level rather than the herd level. This methodological framework enabled the consideration of individual cow factors and ensured statistical power through high sample sizes.

The transition period of each cow served as the basis for the observations. Therefore, a chronological adjustment was implemented for each lactation period in the dataset. This adjustment ensured that events occurring during the dry period were aligned with data from the subsequent lactation period. Completeness of observations was sought by excluding observations without available health records, reproductive information, or survey data. Furthermore, in cases where vaccination protocols were temporarily changed, a buffer period of one month (spanning 15 days before and 15 days after the date of change) was implemented to minimize the risk of misclassification. Particular attention was paid to the herd-specific timing of vaccination to ensure the accurate matching of vaccination periods with corresponding calving dates.

Two of the farms each maintained two herds with slightly different management practices; therefore, for this study, the 20 farms were further divided into 22 herds. Initially, data were made available for 73,378 dairy cows from 22 herds on 20 farms. After data consolidation, 53,370 dairy cows from 21 herds on 19 farms were included in the analysis, representing a total of 120,394 transition periods.

Prepartum vaccinations were administered using three different vaccines: RC (Bovilis^®^ Rotavec^®^ Corona, Intervet, Unterschleißheim, Germany), containing inactivated bovine rotavirus (serotype G6 P5), inactivated bovine coronavirus (strain Mebus), and *E. coli* (K99 Antigen), adjuvanted with mineral oil and aluminum hydroxide (*n* = 27,769); SG (Scourguard^®^ 3, Zoetis, Berlin, Germany), containing live attenuated bovine rotavirus (strain Lincoln), live attenuated bovine coronavirus (strain Hansen), and *E. coli* (K99 Antigen), adjuvanted with Alhydrogel (*n* = 8352); and BS (Bovigen^®^ Scour, Forte Healthcare, Dublin, Ireland), containing inactivated bovine rotavirus (serotype G6 P1), inactivated bovine coronavirus (strain C-197), and *E. coli* (K99 Antigen), adjuvanted with Montanide ISA 206 VG (*n* = 8004). RC and BS are classified as non-live vaccines, while SG contains live and non-live components. For the purpose of this study, the cows vaccinated with RC or BS were allocated to the group NON-LIVE, and the cows vaccinated with SG to the group MIXED. When the type of vaccine could not be determined, the cows were allocated to the group UNKNOWN. As recommended by the manufacturer, RC and BS are applicated once, and SG is applicated twice. As the date of the SG booster was not thoroughly documented, the date of the first application was considered the timing of vaccination in the dataset. In 18,453 cases, the vaccine product could not be identified. All observations with other vaccinations than prepartum vaccination against neonatal calf diarrhea were excluded. Appendix A provides an overview of response and predictive variables, definitions, composition, and values.

Due to the inconsistent description of diagnoses in the documentation of the different herds, all acute uterine diseases were summarized under the category retentio metritis complex (RMC). All diagnoses related to inflammatory events in the uterus (retained placenta, metritis, puerperal intoxication, and puerperal septicemia) were included, provided they were documented within 21 days postpartum. Regarding endometritis, the diagnostic categories endometritis catarrhalis, endometritis mucopurulenta, endometritis purulenta, and pyometra were considered, provided they occurred between day 21 and 56 postpartum.

All herds were bred by artificial insemination. To assess fertility, the *56-day non-return rate* (NRR56) was used as response variable. The NRR56 is defined as the proportion of artificially inseminated cows that do not return to estrus and, consequently, are pregnant, as measured at day 56 post-insemination (p.i.). It is considered less biased than other fertility parameters such as the calving interval, because it includes primiparous and non-pregnant cows. It maintains comparability between herds, although the proportion of pregnant cows is usually overestimated [26].

Available cow- and farm management-related variables were selected and examined for influence on the response variables. Especially those, considered as well studied risk factors of uterine diseases [19,27], were included in the analysis. Farm size, access to pasture, flooring, hygiene score for legs and flank [28], first lactation ages were tested as farm-related variables. Stillbirth, parity, time dry, risk of ketosis, multiples, pen change, calf sex, calving season, energy-corrected milk yield on the first day of milk testing (ECM FTD), ECM 305 in the previous lactation, SCC, time to first service and calving interval as parameters related to the status of the cow and calving. Moreover, the RMC and endometritis were incorporated as potential explanatory variables for the respective outcome variables, in accordance with their established role as significant risk factors for uterine disease and subsequent fertility [29,30]. The complete set of variables and their definitions can be found in Appendix A. Further details on the study design, data collection procedures, and variable definitions have been described previously in [8], which reports on the same study framework and includes partially overlapping variables.

### 2.3. Statistical Analyses

Statistical analyses were performed using R version 4.3.1 [31]. An initial descriptive analysis was performed to provide an overview of the data structure, disease prevalence and fertility. The random forest algorithm [32] was applied in order to obtain a ranking of the importance of the influencing variables. All eligible variables were examined for their influence on the response variables *the RMC*, *endometritis,* and the *NRR56* in the univariable generalized linear mixed effects models. Parity was one of the most significant variables, and those influencing variables originating in the previous lactation (*ECM 305* and *calving interval*) were only available in multiparous cows; therefore, multivariable models were divided into primiparous and multiparous. *Herd* and *calving year* were applied as random effects in mixed-effects models and examined as predictors in random forest models. A nested structure of these random effects was the most appropriate formulation based on the evaluations using Akaike’s information criterion. By applying the function $f\left(x\right)=\frac{0.05}{\sqrt{\frac{x}{100}}}$ [33] to each model, where *x* = number of observations without missing values, an adapted significance threshold was created, to reduce the risk of false discoveries (type I error) due to high numbers of observations in the models. All variables that were significant according to the adapted threshold in the univariable analysis were further included in the subsequent multivariable analysis. Manual backward selection was performed on variables below the adapted *p*-value threshold after multivariable analysis, while constantly comparing model performance. If Akaike’s information criterion was two or more points lower, the variable was excluded. Multicollinearity was assessed by calculating the variance inflation factor.

## 3. Results

### 3.1. Vaccine Type and Timing of Vaccination

Out of the total 120,394 transition period in the dataset, 57,166 transition periods were without vaccination during the dry period (NON-VACC). In 63,228 transition periods, the cows were vaccinated (VACC), of which 35,773 involved a non-live vaccine (NON-LIVE), 8352 a vaccine containing both live and non-live components (MIXED), and 19,103 with an unknown type of vaccine (UNKNOWN). In 30,633 transition periods, vaccination was applied between 6 and 8 weeks before the expected calving date (EARLY); in 14,169 transition periods, vaccination was applied between 2.5 and 4 weeks before the expected calving date (LATE); and in 18,426 transition periods, vaccination was applied, but the exact date was unknown within the dry period (EARLY or LATE) (Table 1).

### 3.2. Production Metrics in Vaccinated and Non-Vaccinated Cows

Among primiparous cows, the proportion of vaccinated individuals was lower (41.3%) compared to multiparous cows (59.5%) (Table 2). Primiparous cows showed a lower ECM 305 yield and a shorter time to first service compared to multiparous cows. Within the primiparous group, the ECM 305 yield was the same for both non-vaccinated and vaccinated cows (8683 L), with a slight difference in time to first service (73 days for the non-vaccinated and 72 days for the vaccinated cows). In the multiparous group, the non-vaccinated cows had a higher ECM 305 yield (10,968 L) compared to the vaccinated cows (10,371 L), while the time to first service was consistent at 75 days for both vaccinated and non-vaccinated groups.

### 3.3. Milk Yield, Performance, and Herd Management Are Most Relevant for Uterine Health and Fertility

Random forest analysis was performed with all significant predictor variables from the univariable general linear mixed effects regression, including herd and calving year, which were previously applied as random effects. The ranking of variable importance showed that ECM 305 in the previous lactation, ECM FTD, herd, and calving interval were among the most influential variables (Figure 1). Prepartum vaccination is among the least influential predictors. However, time and type of vaccine appeared in all three rankings above prepartum vaccination.

### 3.4. Prepartum Non-Live Vaccination Affects Uterine Health and Fertility

The results of multivariable models in primiparous cows (Table 3) showed that vaccination with a non-live vaccine significantly increased the odds of developing the RMC and endometritis, while significantly decreasing the NRR56, when comparing the NON-LIVE group with the NON-VACC group. The odds ratio of 1.73 suggests that individuals who received the non-live vaccine were 1.73 times more likely to develop the RMC, relative to the non-vaccinated group, after controlling for other variables. However, despite a significant p-value, which in this study might be mostly due to the huge sample size, according to Chen et al. (2010), the effect size from OR = 1.73 is classified as small [34]. Primiparous cows who received the non-live vaccine were 3.04 times more likely to develop endometritis. In turn, primiparous cows vaccinated with a non-live vaccine were 24% less likely (OR = 0.76) to be pregnant on day 56 p.i. compared to non-vaccinated primiparous cows. Comparing the groups MIXED and NON-VACC, as well as MIXED and NON-LIVE however showed no significant associations.

The results of the predictors indicate that cows that experienced stillbirth were 1.88 times more likely to develop the RMC. Furthermore, primiparous cows with dystocia had a 61% higher likelihood of developing the RMC (OR = 1.61) and a 54% higher likelihood of developing endometritis (OR = 1.54). Cows carrying multiples were 3.90 times more likely to develop the RMC. Milk yield (ECM FTD) was associated with a 4% reduction in the odds of developing the RMC (OR = 0.96). The risk of ketosis increased the likelihood of developing the RMC by 33%. Finally, the results for calf sex show that cows giving birth to male calves were 24% more likely to develop the RMC.

Generalized linear mixed effects models were conducted, applying herd and calving year as random effects. Empty fields arose when a variable was either not significant in the corresponding univariable analysis or was eliminated by manual backward selection. The vertical wavy line separates the NRR56 from the other response variables due to an inverse association compared to the RMC and endometritis: while higher RMC and endometritis rates are undesirable, higher NNR56 rates represent better fertility. All variables and definitions are listed in Appendix A.

The results of multivariable models with multiparous cows (Table 4) showed that prepartum vaccination with a non-live vaccine significantly increased the likelihood of endometritis and decreased the NRR56 when comparing the NON-LIVE group with the NON-VACC group. The multiparous cows vaccinated with a non-live vaccine were 5.61 times more likely to develop endometritis compared to the non-vaccinated cows, indicating an association between the non-live vaccine and the development of endometritis, with a medium effect size. The multiparous cows vaccinated with a non-live vaccine were 20% less likely to conceive compared to the non-vaccinated cows. In contrast, the cows vaccinated with a mixed vaccine were 87% more likely to conceive compared to those vaccinated with the non-live vaccine, suggesting that the mixed vaccine might be associated with improved fertility.

The results of the predictors show that cows that had a stillbirth were 2.37 more likely to develop the RMC. Moreover, the presence of dystocia increased the odds of the RMC and endometritis by 37% and 39%, respectively, while also significantly decreasing the NRR56 by 9%. The cows with multiple births had a markedly higher risk of the RMC, being 6.40 times more likely to develop the condition. These cows were also 1.31 times more likely to develop endometritis. The cows with ketosis were 16% more likely to develop the RMC. The cows that gave birth to male calves were 18% more likely to develop the RMC. Risk of ketosis was associated with an increased likelihood of RMC by 16%. Milk yield in the current lactation (ECM FTD) was significantly negatively associated with the development of the RMC, but was not significantly associated with either endometritis or the NRR56. However, milk yield in the previous lactation (ECM 305) was positively associated with the RMC and endometritis and negatively associated with the NRR56. Furthermore, longer time to first service increased the NRR56, while longer calving intervals were associated with lower NRR56. Finally, the cows that developed the RMC were 4.28 times more likely to subsequently develop endometritis. Additionally, the occurrence of the RMC was associated with a 10% reduction in the odds of conception.

Generalized linear mixed effects models were conducted, applying herd and calving year as random effects. Empty fields arose when the variable was either not significant in the corresponding univariable analysis or was eliminated by manual backward selection. The vertical wavy line separates the NRR56 from the other response variables due to an inverse association compared to the RMC and endometritis: while higher RMC and endometritis rates are undesirable, higher NNR56 rates represent better fertility. All variables and definitions are listed in Appendix A.

### 3.5. Timing of Vaccination Affects Fertility in Cows Vaccinated with a Non-Live Vaccine

As the vaccination timing for the MIXED group was not documented for the booster, only the NON-LIVE group (17,574 transition periods from 8268 cows) was included in this analysis. In multivariable generalized linear mixed effects models, significant associations between the timing of vaccination and the NRR56 could be found (Figure 2). The predicted NRR56 was 37.6% for the EARLY group and 46% for the LATE group, with a difference in predicted probabilities of 8.4% (Figure 2), while the odds of conception for the late-vaccinated cows were 42% higher. No significant association with the timing of vaccination was found for the RMC and endometritis.

## 4. Discussion

Innate immune mechanisms play a major role in the peripartum period. Vaccine-induced NSEs, which are mainly based on innate immune mechanisms, are rarely studied in the field of uterine health and reproduction. Here, we investigated whether prepartum vaccination against NCD in pregnant cows is associated with the prevalence of postpartum uterine disease and fertility. This was based on the hypothesis that vaccination-induced mechanisms alter the interaction within the immune system or between cells and thus lead to enhanced resistance towards metritis pathogens and improved fertility. Special emphasis was placed on the timing of vaccination, whether early or late during the dry period, as well as the type of vaccine, whether live or non-live. As timely immune activation plays a role at the interface of innate immune mechanisms and uterine diseases and fertility, the timing of vaccination, the timing of parameter measurement, and the type of vaccine (non-live or live) could be decisive.

In multivariable models, vaccinated cows, particularly those immunized with a non-live vaccine, exhibited significantly higher odds of developing uterine disease and lower odds of conception (Table 3 and Table 4). The cows vaccinated with a live vaccine component differed significantly from non-vaccinated cows only in the NRR56 and in multiparity; all other results in this regard were not statistically significant (Table 4). These findings point toward a possible trend of poorer postpartum uterine outcomes following non-live prepartum vaccination. Usually, beneficial NSEs are found in live and adverse in non-live vaccines [3,4]. Accordingly, our results support this pattern, though favorable outcomes associated with live vaccine components in our study were limited to the NRR56 in multiparous cows. In addition, the effect size of all these predictors were small or very small (except for endometritis in multiparous cows) when interpreted according to the recommendations of Chen, Cohen, and Chen [34]. This suggests a small impact of the observed association. The results of the variable importance ranking must also be taken into account: herd management and the cows’ milk yield have the greatest influence on uterine health and fertility in the postpartum cow. In addition to the small effect sizes regarding uterine health and fertility, the influence of milk yield may overshadow the findings of this study. Moreover, the variable importance ranking showed that the timing of vaccination was more relevant than the simple comparison between vaccinated and non-vaccinated cows (or between vaccine types). Currently, vaccine manufacturers recommend to vaccinate cows against NCD between three and twelve weeks prior to the expected calving date, to allow sufficient time for antibody production against the target pathogens and to ensure their passage into the colostrum immediately after calving [35,36,37]. The relation between the timing of vaccination and the timing of pen changes on the acidogenic diet has recently been shown to affect lying time, metabolic profile, and immunoglobulins wit prepartum vaccination with two non-live vaccines against neonatal calf diarrhea (NCD) and mastitis [14,15]. Here, a beneficial effect of vaccination 4 weeks prepartum, followed by pen changes and the acidogenic diet 3 weeks prepartum, compared to all of these procedures 3 weeks prepartum was attributed to additive stressors. In this study, the use and timing of the acidogenic diet and pen changes could not be fully determined retrospectively. The cows in this study were vaccinated between 2.5 and 8 weeks prior to the expected calving date, with multiparous cows usually vaccinated at dry-off and heifers at regrouping. Therefore, a wide range of vaccination timings could be examined in the respective multivariable analysis. To achieve this, the number of observations in the dataset was restricted to cows that had been vaccinated with a non-live prepartum vaccine, including only those cases where the date of vaccination was clear. The results showed a significant association between the timing of vaccination and fertility, with a very small effect size (Figure 2 and Appendix A). For the outcome variables, the RMC and endometritis, no significant results were found.

Our hypothesis, that vaccination-induced mechanisms lead to enhanced resistance to metritis pathogens and improved fertility, was not supported by the results of this dataset. Nevertheless, associations of small effect sizes could be substantiated. Whether these associations are vaccination-induced cannot be fully assessed due to the retrospective, observational, and non-randomized design of this study [38]. The reproducibility of the results across the different multivariable models (e.g., primiparity vs. multiparity; VACC/NON-VACC vs. NON-LIVE/MIXED/NON-VACC), the high statistical power and the adaptation of the significance thresholds to the large dataset [33] argue in favor of a causal relationship. Finally, the findings align with related research that supports beneficial NSEs after live vaccination and adverse NSEs after non-live vaccination [3,4]. However, in order to confirm a causal relationship, controlled prospective trials are necessary.

The associations observed are unlikely to be related to the specific effects of the antibody response of the vaccine. The NCD vaccine targets pathogens, including *E. coli*, bovine corona virus, and rotavirus. Although coronaviruses and rotaviruses are rarely implicated in the pathogenesis of uterine disorders, it cannot be excluded that the generation of antibodies against *E. coli* may interfere with intrauterine *E. coli*. The genotypic specificity of intrauterine *E. coli* is a current area of research [39], and the presence or absence of bacteria plays a part in uterine diseases, but the role of the immune system is estimated as decisive [24]. Dysregulated immune function around calving is associated with impaired and reduced polymorphonuclear (PMN) leucocytes [20,40]. Conversely, early influx of inflammatory PMN leucocytes into the uterus has been associated with reduced uterine disease during the later stages of uterine involution [20]. Innate immune mechanisms are important not only in the postpartum period, but also in the prepartum period [19]. Postpartum, timing again makes a difference: while puerperal metritis usually occurs within ten days postpartum, endometritis is defined as occurring from day 21 postpartum [24]. Carry-over effects of uterine disease on subsequent fertility have been found, although details of the underlying mechanisms are still unclear [25,41]. While PMN leucocytes are known to upregulate inflammation, little is known about their role in the resolution of inflammation. Pascottini et al. hypothesized that this resolution of inflammation is a key feature of PMN leucocytes in the regulation of uterine disease [20]. In dairy reproduction, Ribeiro et al. found that inflammation prior to breeding reduced fertility and suggested additive negative effects of inflammation from different sources (metabolic, NEB, uterine, and non-uterine diseases) [41]. The interplay between systemic and uterine inflammation is not entirely understood, but there is a type or extent of inflammation that leads to maladaptation of the transition cow [42]. Thus, inflammation and, specifically, its upregulation and downregulation, seems to play a key role in uterine health and fertility. We hypothesize that NSEs interfere with these innate regulatory immune mechanisms, especially by increasing the number and competence of PMN leucocytes [6], and thus affect the dysregulated immune system of the periparturient dairy cow.

Large herd records are a time-tested tool for monitoring health parameters in dairy herds [43]. They allow the detection of even small phenomena because of the high statistical power and the thorough examination of subgroups and variables to control for confounders. At the same time, large datasets—and especially herd records—entail several challenges. The herd data used in this study were assumed to contain inaccuracies and inconsistencies in diagnostic records. This assumption is based on a previous study using data from the same source [44]. Inconsistent definitions of clinical diseases and documentation bias are well-known issues in dairy cattle [45,46]. The consolidation of uterine diagnoses is a common method employed to improve diagnostic accuracy [47]. Given the established significance of retained placenta as the foremost risk factor for metritis and the assumption that retained placenta invariably results in inflammation, leading to acute metritis [19,48], a single variable (RMC) was created to summarize acute uterine diseases. In order to further minimize the documentation bias, a combination of diagnoses and reproduction data was utilized to define the outcome variables. The diagnostic parameters retained placenta, metritis, and endometritis were supplemented with the more robust and measurable parameter, the NRR56. Furthermore, this study’s findings were validated through on-site farm visits, thorough data cleaning, and exploratory data analysis, as well as a confounder check using contingency tables and bivariable analyses. In order to circumvent the occurrence of type 1 errors, characterized by the production of false positive significant results, the likelihood of which is known to be amplified in the context of substantial datasets, a more stringent significance threshold was implemented [33].

The risk factors for uterine diseases have been well studied, and the results of this study align closely with the existing literature on the subject [19,27]. It should be noted that certain parameters were either unavailable or only partially available for the model used in this study, including vulval angle, back fat thickness, and bacterial infection [29,49]. The variables referring to the previous lactation of the cow could only be considered in multiparous cows; thus, the models were separated into primiparous and multiparous. Additional factors may also have had an impact. While our analysis showed differences between non-live and live vaccine components, the role of adjuvants should be examined in future studies. Moreover, the cows in the MIXED group received two doses, whereas the non-live vaccines were administered only once. This raises the possibility that the observed NSEs may reflect differences in vaccination schedules rather than vaccine composition. However, the adverse associations observed for non-live vaccines argue against a simple linear effect of dose number. We therefore interpret these findings as more likely related to vaccine components (live vs. non-live). Nevertheless, the potential influence of booster vaccination on NSE outcomes cannot be ruled out and should be addressed in future studies.

In the models used in this study, the highest influential factors for all response variables in this study were ECM FTD, ECM 305 in the previous lactation, and herd management practices, as demonstrated by the variable importance ranking of the random forest analysis. Literature reports contradicting results regarding the influence of milk yield on reproduction [26,50,51,52]. An increase in milk production due to breeding advances and a concurrent decline in reproduction measures has been observed in dairy herds worldwide. Some researchers have proposed that rising milk yields are a cause of rising infertility. However, others have criticized this conclusion or even found a positive correlation. To address this research question, Rearte et al. [53] conducted a multilevel logistic regression analysis. The findings indicated a significant negative association between milk yield and reproductive performance, with a small effect size. In addition, the findings of this study indicate that, while the significance of ECM 305 on the NRR56 is high, the effect size is very small. Correlations between early milk yield and other variables were observed in both positive and negative directions [53,54,55]. In this study, both previous lactation and current early lactation milk yield were negatively associated with uterine health and fertility. It is noteworthy that a notable discrepancy was observed in the correlation between milk yield in the previous lactation and that of the current early lactation; while the former seemed to increase the risk of uterine diseases, high milk yields in the current early lactation seemed to be an indicator for a good uterine health status. The authors hypothesize that regarding uterine disease, the association may be inverse: if the cow’s uterine health is impaired, it affects the overall performance of the animal, resulting in reduced milk yields. Another limitation of this study is the variation in days in milk (DIM) at the time of milk recordings, as data were collected at monthly intervals within the respective herds. To identify potential biases, the relationship between the average DIM on the day of the milk testing and the average milk performance was investigated. The results demonstrated that there was no significant impact on the outcomes.

## 5. Conclusions

This large-scale study confirmed that management practices and milk yield are the most important factors for the dairy cows’ uterine health and fertility. Prepartum vaccination was associated with a small increase in the probability of retained placenta, metritis and endometritis, and reduced fertility. This finding suggests that non-specific effects of prepartum vaccination on uterine health and fertility may occur in dairy cows. However, the retrospective design of this study precludes any determination of causality. It is hypothesized that the associations demonstrated in this study may not outweigh the positive effects of vaccination against NCD on the health and survival of neonatal calves. Instead, these findings shed light upon the current immunological discourse on the NSEs of vaccinations. Further research is required on the mechanisms of NSEs in the critical periparturient phase of the dairy cow, with particular emphasis on the optimal timing of vaccination and the most suitable vaccine components (live, non-live, adjuvants).

## Figures and Tables

**Figure 1 animals-15-02589-f001:**
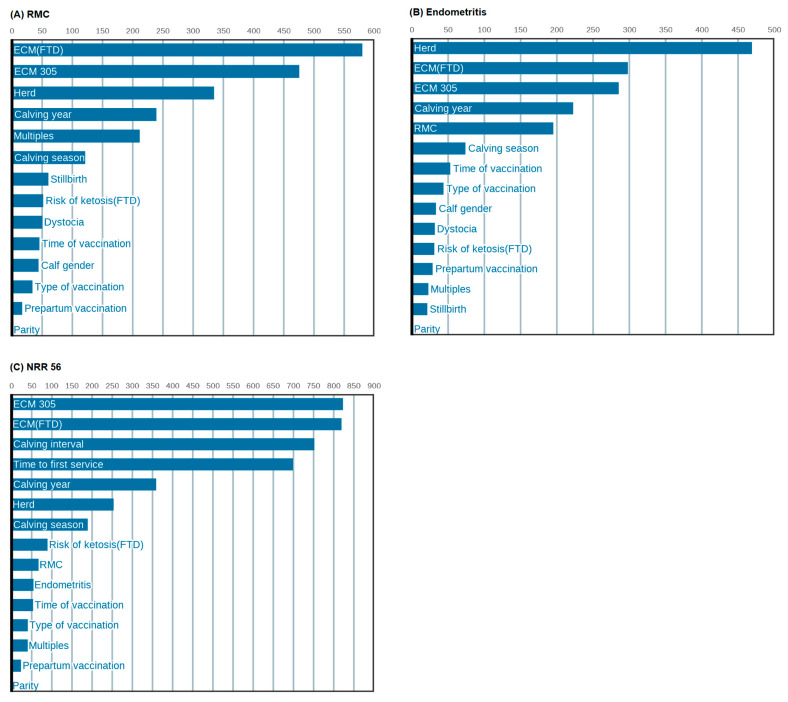
Ranking of importance of influencing variables on the prevalence of the RMC (**A**), endometritis (**B**), and the NRR56 (**C**) by a random forest model. Predicted variable importance is represented by the mean decrease in impurity, which reflects the reduction of uncertainty in the model. Importance reflects the ability of risk factors to accurately predict the response variable (the higher the importance, the better the predictive power).

**Figure 2 animals-15-02589-f002:**
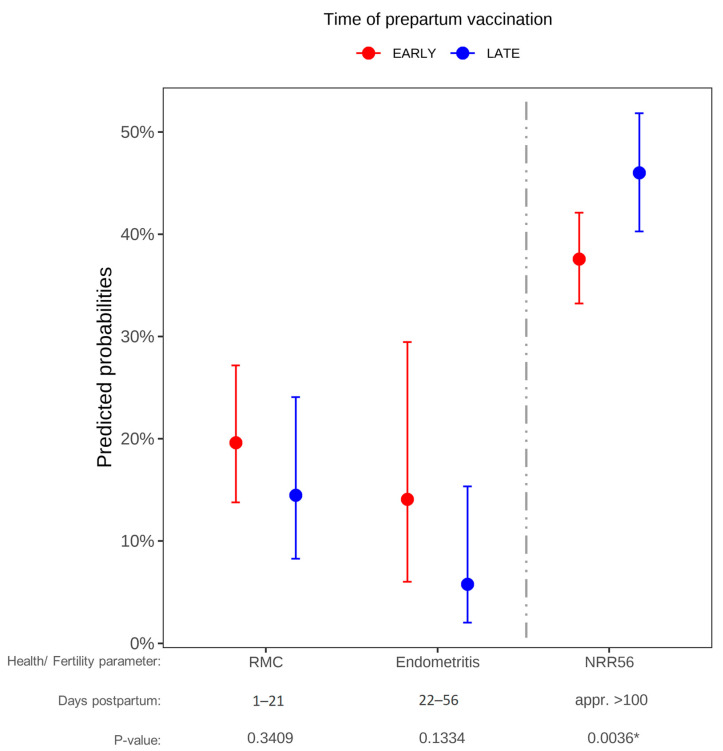
Association between the timing of prepartum vaccination and uterine health and fertility parameters in cows vaccinated with non-live vaccines. Predicted probabilities and 95% confidence intervals are derived from multivariable models. See Appendix A for *p*-values, odds ratios, and confidence intervals for all predictors. *p*-values are marked with * if below the adapted significance threshold, depending on the number of observations, between 0.0051 and 0.0054. The vertical dotted line separates the NRR56 from the other response variables due to an inverse association compared to the RMC and endometritis: while higher RMC and endometritis rates are undesirable, higher NNR56 rates represent better fertility.

**Table 1 animals-15-02589-t001:** Numbers of transition periods by the timing of vaccination and vaccine type.

		Timing of Prepartum Vaccination				
		NON-VACC ^a^	EARLY ^b^	LATE ^c^	EARLY or LATE ^d^	Total
Type of vaccine	NON-VACC ^a^	57,166	0	0	0	57,166
			VACC ^h^ (*n* = 63,228)			
	NON-LIVE ^e^	0	11,238	6336	18,199	35,773
	MIXED ^f^	0	519	7833	0	8352
	UNKNOWN ^g^	0	18,876	0	227	19,103
	Total	57,166	30,633	14,169	18,426	120,394

^a^ No vaccination during the dry period; ^b^ vaccination between 6 and 8 weeks before expected calving date; ^c^ vaccination between 2.5 and 4 weeks before expected calving date; ^d^ vaccination between 2.5 and 8 weeks before expected calving date; ^e^ vaccination with vaccines containing only non-live components; ^f^ vaccination with a vaccine containing both live and non-live components; ^g^ vaccination with unknown components; ^h^ vaccination with any type of vaccine during the dry period.

**Table 2 animals-15-02589-t002:** Means of milk yield and time to first service of prepartum vaccinated or non-vaccinated primiparous and multiparous cows.

	Primiparous Cows		Multiparous Cows	
*n*:	NON-VACC ^a^ 27,081	VACC ^b^ 19,028	NON-VACC ^a^ 30,085	VACC ^b^ 44,200
ECM 305 ^c^	8683	8683	10,968	10,371
Time to first service ^d^	73	72	75	75

^a^ No vaccination during the dry period; ^b^ vaccination during dry period; ^c^ energy-corrected milk yield over 305 days of lactation (kg); ^d^ in days.

**Table 3 animals-15-02589-t003:** Multivariable models: Associations between prepartum vaccine type and uterine health and fertility in primiparous cow.

Postpartum Time Interval	Day 1–21		Day 22–56		Appr. > Day 100	
	RMC ^a^		Endometritis ^b^		NRR 56 ^c^	
	OR ^d^ (95% CI ^e^)	*p*-Value ^f^	OR ^d^ (95% CI ^e^)	*p*-Value ^f^	OR ^d^ (95% CI ^e^)	*p*-Value ^f^
Type of vaccine						
NON-LIVE ^g^/NON-VACC ^h^	1.73 (1.21 to 2.46)	<0.001	3.04 (1.86 to 4.96)	<0.001	0.76 (0.65 to 0.88)	<0.001
MIXED ^i^/NON-VACC ^h^	1.17 (0.34 to 4.05)	0.955	0.14 (0.01 to 2.06)	0.203	0.97 (0.65 to 1.46)	0.988
MIXED ^i^/NON-LIVE ^g^	0.67 (0.20 to 2.28)	0.729	0.05 (0.00 to 0.67)	0.019	1.28 (0.86 to 1.91)	0.305
Stillbirth	1.88 (1.70 to 2.09)	<0.001	1.18 (1.02 to 1.35)	0.024		
Dystocia	1.61 (1.50 to 1.72)	<0.001	1.54 (1.40 to 1.70)	<0.001		
Multiples	3.90 (2.88 to 5.29)	<0.001	1.38 (0.92 to 2.06)	0.114		
ECM (FTD) ^j^	0.96 (0.95 to 0.96)	<0.001	0.98 (0.98 to 0.99)	<0.001		
Risk of ketosis (FTD) ^k^	1.33 (1.23 to 1.44)	<0.001	1.10 (0.99 to 1.22)	0.083		
SCC (FTD) ^l^	1.02 (1.00 to 1.04)	0.074				
Calf sex	1.24 (1.16 to 1.32)	<0.001				
RMC ^a^			3.88 (3.52 to 4.28)	<0.001	0.94 (0.88 to 1.01)	0.075
Endometritis ^b^					0.94 (0.86 to 1.03)	0.205
Calving season ^m^						
spring/autumn	1.09 (0.97 to 1.23)	0.258			0.97 (0.88 to 1.07)	0.854
summer/autumn	1.04 (0.93 to 1.18)	0.783			0.94 (0.85 to 1.04)	0.357
summer/spring	0.96 (0.85 to 1.08)	0.792			0.97 (0.88 to 1.07)	0.842
winter/autumn	1.15 (1.02 to 1.29)	0.018			1.07 (0.97 to 1.17)	0.319
winter/spring	1.05 (0.93 to 1.18)	0.692			1.10 (1.00 to 1.21)	0.057
winter/summer	1.10 (0.98 to 1.23)	0.181			1.13 (1.03 to 1.25)	0.004

^a^ Aggregation of retained placenta and metritis within day 1–21 postpartum; ^b^ endometritis within day 22–56 postpartum; ^c^ non-return-rate refers to day 56 post-insemination; ^d^ odds ratio for pairwise contrasts; ^e^ confidence interval; ^f^
*p*-values are marked in bold, if below the adapted significance threshold of 0.003; ^g^ vaccination during the dry period with a vaccine consisting of only non-live components; ^h^ no vaccination during the dry period; ^i^ vaccination during the dry period with a vaccine consisting of live and non-live components; ^j^ energy-corrected milk yield on the first day of milk testing; ^k^ risk of ketosis was assumed if the fat-to-protein ratio exceeded 1.4, and either the lower limit of protein content (Emin) was undercut or the upper limit of fat content (Fmax) was exceeded on the first day of milk testing. Emin = (4.11 − 0.023 kg milk/day) (1 − 0.35/3.51). Fmax = (5.06 − 0.033 kg milk/day) (1 + 0.68/4.20); ^l^ somatic cell count of the first test day, logarithmically transformed; ^m^ spring (March–May), summer (June–August), autumn (September–November), winter (December–February).

**Table 4 animals-15-02589-t004:** Multivariable models: association between vaccine type and uterine health and fertility in multiparous cows.

Postpartum Time Interval	Day 1–21		Day 22–56		Appr. > Day 100	
	RMC ^a^		Endometritis ^b^		NRR56 ^c^	
	OR ^d^ (95% CI ^e^)	*p*-Value ^f^	OR ^d^ (95% CI ^e^)	*p*-Value ^f^	OR ^d^ (95% CI ^e^)	*p*-Value ^f^
Type of vaccine						
NON-LIVE ^g^/NON-VACC ^h^	1.45 (1.09 to 1.93)	0.007	5.61 (3.47 to 9.09)	<0.001	0.80 (0.71 to 0.91)	<0.001
MIXED ^i^/NON-VACC ^h^	1.45 (0.70 to 3.01)	0.449	1.43 (0.35 to 5.78)	0.822	1.50 (1.02 to 2.22)	0.036
MIXED ^i^/NON-LIVE ^g^	1.00 (0.51 to 1.98)	>0.999	0.25 (0.07 to 0.96)	0.041	1.87 (1.29 to 2.72)	<0.001
Stillbirth	2.37 (2.08 to 2.70)	<0.001	1.15 (0.96 to 1.37)	0.142		
Dystocia	1.37 (1.27 to 1.47)	<0.001	1.39 (1.26 to 1.53)	<0.001	0.91 (0.85 to 0.97)	0.003
Multiples	6.40 (5.69 to 7.22)	<0.001	1.31 (1.11 to 1.54)	0.001		
ECM (FTD) ^j^	0.95 (0.95 to 0.95)	<0.001	1.00 (0.99 to 1.00)	0.275	1.00 (0.99 to 1.00)	0.061
Risk of ketosis (FTD) ^k^	1.16 (1.09 to 1.24)	<0.001	1.13 (1.04 to 1.23)	0.004		
SCC (FTD) ^l^	1.01 (0.99 to 1.02)	0.358				
Calf sex	1.18 (1.12 to 1.26)	<0.001				
RMC ^a^			4.28 (3.92 to 4.67)	<0.001	0.90 (0.84 to 0.96)	0.003
ECM 305 ^m^	1.00 (1.00 to 1.00)	<0.001	1.00 (1.00 to 1.00)	<0.001	1.00 (1.00 to 1.00)	<0.001
Calving interval					1.00 (1.00 to 1.00)	<0.001
Time to first service					1.00 (1.00 to 1.00)	<0.001
Calving season ^n^						
spring/autumn	1.17 (1.05 to 1.31)	0.002			0.89 (0.80 to 0.97)	0.006
summer/autumn	1.10 (0.99 to 1.22)	0.087			0.85 (0.78 to 0.93)	<0.001
summer/spring	0.94 (0.84 to 1.05)	0.452			0.96 (0.87 to 1.06)	0.736
winter/autumn	1.04 (0.93 to 1.15)	0.819			1.05 (0.96 to 1.14)	0.511
winter/spring	1.05 (0.93 to 1.18)	0.692			1.10 (1.00 to 1.21)	0.057
winter/summer	1.10 (0.98 to 1.23)	0.181			1.13 (1.03 to 1.25)	0.004

^a^ Aggregation of retained placenta and metritis within days 1–21 postpartum; ^b^ endometritis within days 22–56 postpartum; ^c^ non-return-rate refers to day 56 post-insemination; ^d^ odds ratio for pairwise contrasts; ^e^ confidence interval; ^f^
*p*-values are marked in bold if below the adapted significance threshold of 0.003; ^g^ vaccination during the dry period with a vaccine consisting of only non-live components; ^h^ no vaccination during the dry period; ^i^ vaccination during the dry period with a vaccine consisting of both live and non-live components; ^j^ energy-corrected milk yield on the first day of milk testing; ^k^ risk of ketosis was assumed if the fat-to-protein ratio exceeded 1.4, and either the lower limit of protein content (Emin) was undercut or the upper limit of fat content (Fmax) was exceeded on the first day of milk testing. Emin = (4.11 − 0.023 kg milk/day) (1 − 0.35/3.51). Fmax = (5.06 − 0.033 kg milk/day) (1 + 0.68/4.20); ^l^ somatic cell count of the first test day, logarithmically transformed; ^m^ energy-corrected milk yield over 305 days in the previous lactation; ^n^ spring (March-May), summer (June–August), autumn (September–November), winter (December–February).

## Data Availability

The data generated and analyzed during this study are not publicly available due to reasons of confidentiality. Anonymized data are available upon reasonable request.

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
