# Peer review of "Non-Specific Effects of Prepartum Vaccination on Uterine Health and Fertility: A Retrospective Study on Periparturient Dairy Cows"

_animals, 2025, doi:10.3390/ani15172589_

Round 1
Reviewer 1 Report
Comments and Suggestions for Authors
This manuscript is well written and addresses an interesting topic on cattle vaccination, specifically the non-specific effects (NSE) of vaccination during the prepartum period on uterine health and fertility. The study analyzes thousands of records from 20 farms in Germany. The retrospective analysis compares farms that use non-live vaccines against neonatal calf diarrhea with those that do not vaccinate. The statistical methods suggest potential negative NSE associated with non-live vaccines, which is consistent with previous reports.
I have some minor comments that should be considered:
-
Line 62: The first time the acronym NCD appears, its meaning should be included.
-
Lines 86–87: Please revise the wording for clarity.
-
Line 109: Please clarify whether the on-farm survey was applied to all farms.
-
Line 156: E. coli should be italicized. Please correct throughout the text.
-
Line 185: The acronym ECM FTD is not explained anywhere in the text. Please provide clarification.
Reviewer 2 Report
Comments and Suggestions for Authors
This reviewer accepted the MS after a minor revision. The authors are encouraged to address the comments and concerns made in the attached comments to the authors.
